# Characterization of Persistent Uncontrolled Asthma Symptoms in Community Members Exposed to World Trade Center Dust and Fumes

**DOI:** 10.3390/ijerph17186645

**Published:** 2020-09-11

**Authors:** Joan Reibman, Caralee Caplan-Shaw, Yinxiang Wu, Mengling Liu, Milan R. Amin, Kenneth I. Berger, Maria L. Cotrina-Vidal, Angeliki Kazeros, Nedim Durmus, Maria-Elena Fernandez-Beros, Roberta M. Goldring, Rebecca Rosen, Yongzhao Shao

**Affiliations:** 1Department of Medicine, NYU Grossman School of Medicine, New York, NY 10016, USA; caralee.caplan-shaw@nyulangone.org (C.C.-S.); kenneth.berger@nyulangone.org (K.I.B.); angeliki.kazeros@nyulangone.org (A.K.); Nedim.Durmus@nyulangone.org (N.D.); MariaElena.Fernandez-Beros@nyulangone.org (M.-E.F.-B.); roberta.goldring@nyulangone.org (R.M.G.); 2Department of Population Health, NYU Grossman School of Medicine, New York, NY 10016, USA; Yinxiang.Wu@nyumc.org (Y.W.); mengling.liu@nyulangone.org (M.L.); Yongzhao.Shao@nyulangone.org (Y.S.); 3Department of Environmental Medicine, NYU Grossman School of Medicine, New York, NY 10016, USA; 4World Trade Center Environmental Health Center, NYC H+HC, New York, NY 10016, USA; maria.cotrina@nyulangone.org (M.L.C.-V.); rebecca.rosen@nychhc.org (R.R.); 5Department of Otolaryngology-Head and Neck Surgery, NYU Grossman School of Medicine, New York, NY 10016, USA; milan.amin@nyulangone.org; 6Department of Psychiatry, NYU Grossman School of Medicine, New York, NY 10016, USA

**Keywords:** World Trade Center, WTC survivors, environmental exposure, asthma, rhinosinusitis, bronchial hyper-responsiveness, paradoxical vocal fold movement

## Abstract

The destruction of the World Trade Center (WTC) towers on the 11th of September, 2001 released a vast amount of aerosolized dust and smoke resulting in acute and chronic exposures to community members as well as responders. The WTC Environmental Health Center (WTC EHC) is a surveillance and treatment program for a diverse population of community members, including local residents and local workers with WTC dust exposure. Many of these patients have reported persistent lower respiratory symptoms (LRS) despite treatment for presumed asthma. Our goal was to identify conditions associated with persistent uncontrolled LRS despite standard asthma management. We recruited 60 patients who were uncontrolled at enrollment and, after a three-month run-in period on high-dose inhaled corticosteroid and long acting bronchodilator, reassessed their status as Uncontrolled or Controlled based on a score from the Asthma Control Test (ACT). Despite this treatment, only 11 participants (18%) gained Controlled status as defined by the ACT. We compared conditions associated with Uncontrolled and Controlled status. Those with Uncontrolled symptoms had higher rates of upper airway symptoms. Many patients had persistent bronchial hyper-reactivity (BHR) and upper airway hyper-reactivity as measured by paradoxical vocal fold movement (PVFM). We found a significant increasing trend in the percentage of Controlled with respect to the presence of BHR and PVFM. We were unable to identify significant differences in lung function or inflammatory markers in this small group. Our findings suggest persistent upper and lower airway hyper-reactivity that may respond to standard asthma treatment, whereas others with persistent LRS necessitate additional diagnostic evaluation, including a focus on the upper airway.

## 1. Introduction

We, and others, have reported the onset of respiratory symptoms after the destruction of the World Trade Center (WTC) towers on 9/11/2001 in community members: those who lived (residents), worked (local workers), attended schools (students), or were passing by (passerby). These individuals had potential for acute exposures from the dust clouds created as the WTC towers fell on 9/11/2001, and/or chronic exposures from resuspended dust and fumes from the fires over subsequent months [1,2,3,4,5]. The massive amounts of large and small particles, present for months in outdoor and indoor areas, were notable for their ability to be easily resuspended, their extremely basic pH (pH 9–10), and the complex mixture of indoor and outdoor toxins [6,7,8]. Knowledge of these components has contributed to the recognition of the development of lower respiratory symptoms (LRS). In view of reports of the biologic plausibility and reports of airway hyper-responsiveness in exposed populations [9,10], these symptoms were considered consistent with irritant-induced asthma when no other cause could be identified [11].

The WTC Environmental Health Center (WTC EHC) is a treatment program developed for community members with physical and mental health symptoms, which developed after exposures to the WTC dust and fumes [3,5]. Since many of these individuals with LRS were considered to have irritant-induced asthma, a protocol was implemented early in the development of the program for treatment of these patients according to the asthma guidelines of the times [12,13]. These guidelines included step-up therapy with inhaled corticosteroids (ICS) and eventual addition of long acting bronchodilators (LABAs) to gain control of symptoms. Despite this standardized approach, many patients with WTC exposures continued to report persistent uncontrolled LRS [14,15,16,17]. Many had spirometry values that were within normal limits despite their continued symptoms. Uncontrolled asthma is predominantly attributed to poor adherence to treatment regimens and the mainstay of therapy includes the reinforcement of the use of medications such as ICS and LABAs [18]. Since the continued use of high-dose ICS has potential for adverse effects, it is imperative to understand whether these agents remain appropriate therapy for these patients. The aim of this study was to identify conditions associated with uncontrolled LRS in patients who were adherent to high-dose ICS/LABA. After enrolling patients and monitoring adherence to medications, we analyzed risks for persistent uncontrolled LRS and measured the association of lung function, including small airway abnormalities, upper respiratory symptoms, persistent airway hyper-responsiveness, including bronchial hyper-responsiveness and paradoxical vocal fold motion (PVFM), and inflammatory markers. Few patients attained Controlled status, and our findings suggested that that there was a complex pattern of responses with an important role for upper airway symptoms and airway hyper-responsiveness with LRS.

## 2. Methods

### 2.1. Study Design

We performed a one-arm intervention study to identify persistent uncontrolled LRS in patients receiving standard treatment for asthma. We planned a single group study, in which all subjects received a single intervention and the outcomes were assessed over time, because of the strong therapy preference with ICS/LABA for treatment for these symptoms [19]. After identification of patients with reported uncontrolled asthma and normal spirometry in the WTC EHC, we performed a three-month monitored treatment with combined high-dose ICS and LABA and measured asthma symptom control after this intervention as defined by the Asthma Control Test (ACT) as our primary comparator [20]. Our goal was to perform subgroup analyses to identify patients that obtained symptom control (ACT ≥ 20) (Controlled) and compare these patients with those that remained uncontrolled (ACT < 20) (Uncontrolled) to identify rates of associated conditions between these two groups to assess whether these conditions were predictors of the health outcome (e.g., control). Conditions under study included persistent lung function abnormalities including measures of small airway function, upper airway symptoms, airway hyper-responsiveness, including measures of bronchial hyper-responsiveness (BHR) and laryngeal dysfunction (paradoxical vocal fold movement; PVFM), and exhaled and blood inflammatory markers. We report on this original study group as those who we enrolled as “uncontrolled” at V1. We performed an interim analysis during the study implementation and identified that only a few patients had become Controlled at the final study visit (Visit 4; V4), raising concern about an appropriately sized Controlled group for the final analysis. Due to this, we amended our study procedure to enrolled patients who had a history of LRS but were controlled at the first study visit (V1) to expand our final Controlled population. We report on this total group separately as those who were enrolled as controlled or uncontrolled at V1. A diagram of this study design and subsequent modification is shown (Figure 1).

### 2.2. Study Participants

The WTC EHC enrolls community members with physical or mental health symptoms caused or aggravated by their WTC exposures [3,5]. The association between WTC exposures and symptoms is defined by exposure and temporal occurrence of symptoms [21]. Patients undergo a standardized initial visit and subsequent monitoring visits (MV), which include respiratory assessments including the ACT and spirometry [3,5]. Patients with LRS consistent with asthma are treated according to published asthma guidelines and their updates [13]. For this study, patients were identified by an initial review of their symptoms and ACT scores obtained during their MV. A chart review was performed to review inclusion and exclusion criteria. Patients who fit criteria by chart review were approached by telephone or in person in the WTC EHC to assess their interest in study participation.

WTC EHC patients were included in the study if they were ≥18, had LRS at their standardized WTC EHC initial visit, and had been prescribed a controller to treat their LRS including, but not limited to, an ICS. They were subsequently required to have uncontrolled LRS with an ACT score < 20 at a subsequent standardized WTC EHC monitoring visit as well as spirometry values within the normal limit (WNL) [22]. Patients were excluded from the study if they reported a >5 pack-year history of tobacco use, cardiac disease, active cancer, other known lung diseases (sarcoidosis, ILD, bronchiectasis), asthma predating 9/11, spirometry at a WTC EHC monitoring visit that was below normal limits, or an abnormal CXR or chest computerized tomography. Based on an interim analysis, we subsequently amended the inclusion criteria to include patients who met all other criteria, but who had an ACT > 20 at a WTC EHC monitoring visit (controlled) to enrich the controlled group to allow for additional analysis of our associated conditions. The study was approved by the New York University School of Medicine Institutional Review Board (IRB) and all patients signed consent (IRB numbers i06-1 and S13-00448).

### 2.3. Study Procedure

Patients underwent a first study visit (V1) to confirm a history of LRS, normal spirometry values, and their clinical treatment status with a prescribed ICS or ICS/LABA. Patients subsequently underwent standardized questionnaires to confirm their asthma control status (ACT). Patients with an ACT < 20 at their first study visit were placed on high-dose ICS/LABA (Fluticasone propionate 500 mcg/salmeterol 50 mcg inhalational powder, 1 puff twice daily) with albuterol MDI as needed for rescue. Many patients could not tolerate this delivery system, and treatment was modified to include fluticasone propionate 230 mcg/salmeterol 21 mcg HFA aerosol inhaler 2 puffs twice daily with a large volume spacer, a treatment modality that was better tolerated. Patients were monitored monthly for 3 months for adherence at which time they underwent their final study visits. At the final study visit (V4), they completed a final ACT assessment for classification as “Controlled” (ACT ≥ 20) or “Uncontrolled” (ACT < 20).

### 2.4. Measurements

Standardized assessment of LRS included the Asthma Control Test (ACT) and the Asthma Symptom Utility Index (ASUI) [20,23,24]. Assessment of respiratory functional status was performed with the 5-question modified Medical Research Council (mMRC), which reports perceived respiratory disability [25], and a 6 min walk test (6MWT) [26]. Upper respiratory symptoms consistent with sinus symptoms, cough, and voice functioning were assessed with the ICSD Sinus Symptom Score [27,28], the Leicester Cough Questionnaire (LCQ) [29] and the Voice Handicap Index (VHI) [30]. Lung function was measured by spirometry performed according to standard ATS/ERS guidelines [31]. Predicted values for spirometry measures were derived from NHANES III [22] and abnormal spirometry was defined by FEV_1_, FVC, or FEV_1_/FVC measurements below the lower limit of normal (LLN) [32]. A positive BD response for spirometry was defined according to ATS/ERS guidelines after inhalation of 180 mcg albuterol sulfate delivered via a large volume spacer [32].

Small airway function was measured using forced oscillation techniques before and after BD administration. Forced oscillation techniques were performed using impulse oscillometry (IOS) under tidal breathing via the Jaeger Impulse oscillation system^®^. Measurements included respiratory resistance at 5Hz (R_5_) and the difference in resistance from 5 to 20Hz (R_5–20_) as an index of frequency dependence of resistance (FDR), with higher values considered more abnormal. Upper limits of normal (ULN) for R_5_ and R_5–20_ (3.96 and 0.76 cmH_2_O/L/s, respectively) were based on previous reports of asymptomatic non-smoking subjects with normal spirometry [33,34,35]. A positive BD response for IOS measurements was defined as a decrease in R_5_ of ≥1.40 cm H_2_O/L/s, based on the 95th percentile for BD response in healthy adults in normative data on forced oscillation [34,35].

The presence of persistent bronchial hyper-responsiveness (BHR) despite treatment was assessed using methacholine challenge testing according to ATS recommendations [36]. We used a 2 min tidal breathing exposure from a nebulizer up to a maximum dose of 16 mg/mL. The degree of bronchial hyper-reactivity (BHR) was determined by the calculated provocative concentration that resulted in a 20% fall in FEV_1_ (PC_20_). Since patients were on treatment, we defined BHR as a PC_20_ < 16mg/mL. Subjects withheld all inhaled medications on the morning of the study.

Analysis for paradoxical vocal fold movement was performed during the final visit at the NYU Voice Center. Flexible nasolaryngoscopy was performed using a Pentax 3.3 mm distal chip nasolaryngoscope (Pentax Medical, Tokyo, Japan). The flexible endoscope was inserted through the nasal cavity and was used to perform a comprehensive examination of the nasal cavity, pharynx, and larynx. Minimal anesthesia was used to avoid altering the response to stimuli. The larynx was observed during quiet respiration and modal phonation with steady light and stroboscopy. Presence of discharge, mucosal inflammation, and nasal polyps was noted. Following anatomic assessment, the endoscope was positioned above the vocal folds for a functional assessment. The larynx was observed during quiet respiration, modal phonation, and during specified vocal tasks designed to elicit maximal vocal fold excursion. Odor challenges were conducted as per Forrest et al. [37,38]. If no abnormal motion was identified, the patient was directed to participate in activities of exertion until symptomatic (stair climbing, stationary bike riding) with re-examination of the larynx to observe for functional narrowing of the glottis. The presence of paradoxical vocal fold motion at rest or with provocation was noted.

### 2.5. Statistical Analysis

Mean and standard deviation (SD) or median and inter-quartile range (IQR) were used to summarize continuous variables. Counts and proportions were used to describe categorical variables. To compare variables between the final Controlled and Uncontrolled groups, Wilcoxon rank test and Fisher exact test were used for continuous and categorical variables, respectively. The exact version of the Cochran–Armitage test for trend (CATT) was used to assess significance of the increasing trend among proportions of Controlled with respect to the number of positives in PVFM, AHR. Statistical analyses were conducted using R (R Core Team, 2013, Vienna, Austria).

## 3. Results

### 3.1. Patient Recruitment

A final group of 109 patients agreed to participate in the study. Of these, 19 patients failed a subsequent screening and 17 failed to complete the study. Of these total patients, 60 fit criteria as uncontrolled at V1 (ACT < 20) (uncontrolled population) and were considered the primary study population as originally defined. Thirteen patients were recruited despite being controlled at V1 for a subsequent analysis of the total studied population of 73 patients, which included patients who were enrolled as controlled at V1. Among the total group of 73 study patients, 54 were Uncontrolled by the final visit (V4) and 19 (26%) were Controlled.

Demographic, clinical, and exposure characteristics of the primary study population of patients who were uncontrolled at V1 (*n* = 60) are shown in Table 1. Overall, the patients were predominantly women (70%), with a median age of 57 (range 48–62). Participants were of diverse race/ethnicity with most self-reporting as Hispanic. The median BMI was 29. Over 50% reported exposure to the WTC dust clouds, and most had WTC exposure as local workers. As defined by the status at the final study visit, the Uncontrolled group included 49 participants (82%) and 11 participants were included in the Controlled group (18%). There was a slight difference in the distribution of race/ethnicity, but no other differences in patient characteristics or WTC exposure measures were identified between Uncontrolled or Controlled participants. We examined findings in our larger group of patients who were enrolled as either uncontrolled or controlled at V1 as per our expanded study design (Appendix A). At the final study visit, this group included 54 participants who were Uncontrolled and 19 participants who were Controlled. No significant demographic or clinical characteristic differences were identified between Uncontrolled or Controlled in this larger group.

To corroborate the presence of LRS and functional impairment at this visit, patients also completed the Asthma Symptom Utility Index and the mMRC. The ASUI is a validated 11-question instrument that assesses frequency and severity of asthma symptoms and has demonstrated reliability, construct validity, and responsiveness to change [23,24]. Higher scores are consistent with milder symptoms. As shown in Table 1, there was a significant difference in ASUI scores in Uncontrolled compared with Controlled participants (median of 0.6 vs. 0.8, *p*-value = 0.001), consistent with our classification of symptoms using the ACT. There was a suggestion that more Uncontrolled participants scored higher on the mMRC compared with Controlled (20 vs. 9% with an mMRC ≥ 3), although this did not reach statistical significance. Both Uncontrolled and Controlled participants had reduced 6MWT when compared to values reported for control patients in epidemiologic studies [26,39]. These findings were similar when we analyzed patients in the larger group (Appendix A). These data suggested that the Uncontrolled and Controlled patients were distinguished by differences in lower respiratory symptoms using multiple instruments.

### 3.2. Upper Respiratory Symptoms

Dyspnea, wheeze, and cough are LRS symptoms that reduce ACT scores, however, these are non-specific symptoms that may also reflect upper airway abnormalities. We therefore also assessed upper respiratory symptoms in the primary study group and used multiple instruments (Table 2).

For rhinosinusitis symptoms, we used the ICSD scoring system [27,28], a likert scale of 1–10 for each symptom, with higher scores reflecting more symptoms. A mean score of 8.8 is reported in normal populations and patients with a confirmed rhinosinusitis score in the mid 30 s [28]. Uncontrolled participants were more symptomatic (median score 35, IQR 24.0–42.5) compared with Controlled (median score 26, *p* < 0.03). We used the Leicester Cough Questionnaire (LCQ), a cough-specific health status questionnaire that assesses the physical, psychological, and social domains of cough. A higher score is associated with better health and a minimally important difference has been reported to range between 1.3 and 2.56 [40]. We identified a lower score in Uncontrolled participants compared to Controlled (*p* < 0.05). We used the Voice Handicap Index 10 [30,41] as a measure of laryngeal dysfunction or hypersensitivity to capture the “overall” state of voice handicap. The normative value for this instrument is 2.83, with a score > 11 considered abnormal [30,41]. The median value for both Uncontrolled and Controlled patients was below that considered abnormal and no significant difference was noted for voice handicap between the two groups. These findings were similar when examined in the larger group of patients enrolled at V1 with uncontrolled and controlled symptoms (Appendix A), reinforcing these findings. In addition, these findings persisted even when analyzed for those with more uncontrolled symptoms (ACT < 15, n = 29) and Controlled (data not shown).

### 3.3. Lung Function: Spirometry and Small Airway Measures

As defined by the study protocol, patients were included with normal spirometry values. As shown for the final study visit (Table 3), median values for all spirometry measures remained within normal limits for Uncontrolled and Controlled patients with no difference detected between participants for any measure of pre- and post-BD spirometry.

We have previously shown an association between small airway abnormalities and persistent LRS in WTC-exposed community members [15,42,43,44]. In addition, the presence of small airway abnormalities is increasingly being recognized to contribute to LRS in patients with asthma [45]. We therefore examined small airway function using FOT in Uncontrolled vs. Controlled patients (Table 3). We used a normal value of R_5_ as < 3.96 cmH_2_O/L/s [33,34,35] and we considered a normal value of R_5–20_ to be < 0.76 cmH_2_O/L/s based on previously published data [33,34,35]. There was a suggestion of an elevated median pre- and post-BD R_5–20_ in Uncontrolled participants compared with Controlled, although this did not reach statistical significance. Results were similar when we analyzed the larger group of patients who were enrolled as uncontrolled or controlled at V1 (Appendix A). To assess whether there were differences between patients with extremes of control symptoms, we compared patients with more uncontrolled symptoms (ACT < 15, n = 29) [20] with Controlled (ACT > 20). We were unable to detect significant differences in lung function between these two groups (Appendix A).

### 3.4. Airway Hyper-Responsiveness

Due to the possibility that irritants have the potential to cause hyper-reactivity of the whole airway and that the symptoms reported in the ACT could reflect upper as well as lower airway conditions, we measured airway hyper-responsiveness using measures of bronchial hyper-reactivity (BHR) with a methacholine challenge test, and upper airway reactivity with an examination for paradoxical vocal fold movement (PVFM).

Many patients with LRS associated with WTC dust/fume exposure are considered to have irritant-induced asthma and many, although not all, have been demonstrated to have BHR as measured by a methacholine challenge test [4,10,46]. Since most of the patients in the WTC EHC were not assessed for BHR before starting therapy, we assessed whether persistent BHR while on therapy was associated with uncontrolled LRS and performed methacholine challenge testing while participants were receiving treatment with high-dose ICS/LABA. Since patients were on ICS/LABA therapy, we defined BHR at or below 16 mg/mL methacholine as a positive response. BHR was identified in 50% of tested patients. Table 4 displays the analysis of negative or positive BHR as a predictor for Controlled status. Among those with BHR, 25% were Controlled, while among those without BHR, only 12.5% were Controlled, suggesting the possibility that BHR was associated with a higher percentage of Controlled.

To further evaluate upper airway function as a possible cause for symptoms, we performed direct laryngoscopic evaluation. Direct laryngoscopic visualization of the larynx is considered the “gold standard” for assessing laryngeal dysfunction, including assessment for paradoxical vocal fold movement (PVFM) or functional narrowing of the laryngeal inlet. An additional assessment includes challenge with odors or exercise, since many patients may not display paradoxical movement until provoked. Accordingly, we performed laryngoscopic visualization at rest or after provocation with various odors or exercise. Few participants (4 patients in each group) had unprovoked paradoxical narrowing of the laryngeal inlet, whereas many patients had provoked PVFM. Of those measured, 48% had any PVFM (provoked or unprovoked). We then used the presence of any PVFM as a predictor of Controlled status (Table 4). Among those with any PVFM, 22% were Controlled, while among those without PVFM, only 17% were Controlled, suggesting the possibility that PVFM was associated with a higher percentage of Controlled.

Since asthma and PVFM can co-exist, we also evaluated the concurrence of both BHR and PVFM. Most participants (86%) had either BHR, PVFM, or both. Using the combination of BHR and PVFM as a predictor, among those positive for both, the percentage of Controlled was 33%, while among those negative for both, the percentage for Controlled was 0%. Among those with one positive predictor (either BHR or PVFM), the percentage of Controlled was 21%, a value in between those for both positive or both negative. The trend in proportions was statistically significant using the exact Cochran–Armitage test for trend (CATT_exact, *p* = 0.02). The same significant trend on the percentage of Controlled with respect to the number of positives in (PVFM, AHR) was also observed when we analyzed the patients recruited with both uncontrolled and controlled symptoms at V1 (Appendix A).

### 3.5. Inflammatory Biomarkers

Asthma is a heterogeneous process with differences in underlying inflammation that can impact treatment [47]. Clinically measurable biomarkers include exhaled nitric oxide (FeNO), blood eosinophils, and IgE, all associated with type 2 inflammation (T2) [47]. Levels of these markers have been reported to be variable in studies of WTC-exposed responders and community members and to be associated with upper and lower airway disease [48,49,50]. Since corticosteroid-resistant asthma can also be associated with non-type 2 inflammation including neutrophilic asthma, we analyzed blood markers of inflammation. As shown (Table 5), FeNO levels were low in the whole study group (16.5 ppb) and although a statistically significant difference was detected between Uncontrolled and Controlled patients, the difference was slight (18.8 ppb [14.0, 22.0] vs. 13.0 [10.2, 13.9] respectively, median [IQR] *p* = 0.01). Similarly, absolute blood eosinophil counts were low in Uncontrolled vs. Controlled groups. Median total IgE levels were also low at 41.0 [20.0, 135] vs. 31.0 [21/5, 57] in Uncontrolled and Controlled, respectively, and did not differ between the two groups, with 47 and 27% having any allergen-specific IgE. To evaluate for differences in blood inflammatory markers, we analyzed total white blood cell counts, eosinophils, neutrophils, lymphocytes, and their ratios [51]. We did not detect a difference in % neutrophils, or blood neutrophil lymphocyte ratio (NLR), eosinophil lymphocyte ratio (ELR), or eosinophil neutrophil ratio (ENR). Thus, we were unable to detect obvious markers of inflammation in the blood that distinguished between Uncontrolled and Controlled participants. Findings were similar even when analyzed for those with more uncontrolled symptoms (ACT < 15, n = 29) or in total recruited study population (data not shown).

## 4. Discussion

WTC-exposed patients with toxic inhalation often report symptoms of shortness of breath, chest tightness, and wheezing. These patients have been considered to have non-occupationally related irritant-induced asthma as defined in numerous reviews [11,52,53]. As such, they have been treated with guideline-based management for asthma with step-up therapy resulting in high-dose ICS and LABA and control of symptoms measured using standard instruments. Although many WTC-exposed patients reported some improvement in symptoms, others have had an incomplete response, with persistent lower respiratory symptoms, often in the setting of preserved lung function [5,15,43]. Poor adherence to medical regimens is usually considered the main reason for poor control of asthma symptoms, however additional explanations can include corticosteroid-insensitive pathways of airway hyper-responsiveness, small airway involvement not accessed by current inhalers, or symptoms due to alternative or co-morbid processes including those involving the upper airway. This study was designed to identify potential mechanisms for continued LRS despite adherence to high-dose treatment for asthma (high-dose ICS/LABA). Surprisingly, although we monitored medication use closely for the three-month study period, we failed to control symptoms in most participants, necessitating a search for alternative explanations for persistent LRS other than adherence. Comparison of Uncontrolled and Controlled participants revealed a significant potential contribution of upper airway symptoms in the continued reporting of LRS. In addition, there were high rates of BHR and PVFM and we surprisingly, we found a significant increasing trend in the percentage of Controlled with respect to the presence of BHR and PVFM. Our study suggests multiple potential mechanisms for Uncontrolled LRS symptoms.

Poor asthma control is usually ascribed to poor adherence with prescribed medications [14,54,55,56,57] and we assumed that many of the persistent symptoms in the WTC EHC patients would be due to failure to use appropriate medications. We therefore presumed that LRS would improve with adherence to a treatment of ICS/LABA and designed a run-in period to confirm adherence before defining study status as Uncontrolled or Controlled. Our interim analysis suggested that few patients gained Controlled status despite this observational period and indeed, by the end of the study period, only 11 participants, 18% of the 60 patients that were uncontrolled at V1, reached Controlled status. This finding reinforced the need to search for alternative explanations of LRS. The findings are also consistent with a recent study in the WTC Health Registry, in which adherence with asthma therapy was associated with decreased control of symptoms [14].

We used the ACT to define Uncontrolled and Controlled participants. The ACT is a self-administered instrument with a composite score that has been validated against specialists’ rating of control, spirometry, and quality of life [20,58]. We further confirmed the presence of LRS with an additional assessment using the ASUI, providing support for the symptom reports. Despite the presence of increased symptoms in the Uncontrolled participants, we were unable to detect a clear difference in functional impairment between the two groups as measured by the mMRC or the 6MWT. Importantly, most participants had reduced 6MWT when compared to historical normal groups [39]. The failure to identify differences between the two groups may be due to the small study population or possible discordance between the ACT and functional impairment as measured by the mMRC and the 6MWT. In addition, all our patients had a history of reported symptoms and there might be other causes of functional impairment despite a composite score consistent with ACT.

Dyspnea, wheeze, and cough are clinical hallmarks of upper airway as well as lower airway disease. We used the ACT to measure lower airway symptoms consistent with asthma, but the possibility exists that these symptoms may have been due to upper airway symptoms from rhinosinusitis and/or laryngeal dysfunction, all of which are well-described in WTC-exposed patients. Rhinosinusitis symptoms assessed with two instruments (ICDS, Leicester Cough Questionnaire) differed between the Uncontrolled and Controlled groups, with more symptoms in the Uncontrolled group. This important finding suggests that upper airway mechanisms contributed to the reported lower airway symptoms. The finding highlights an important need to identify and focus on modifying upper airway disease that may be concurrent with lower airway symptoms or disease and raises an important question about reassessing treatment.

Irritant-induced lung disease is associated with hyper-responsiveness which may manifest as lower airway bronchial hyper-responsiveness or laryngeal hypersensitivity. We therefore assessed overall airway hyper-responsiveness using measures of upper and lower airway hyper-responsiveness and assessed both BHR and PVFM. Although the diagnosis of asthma relies on the presence of BHR [59,60], due to clinical constraints, few WTC EHC patients underwent BHR measurements before initiation of clinical treatment. However, numerous studies have described BHR in many exposed to WTC dust and fumes [4,10,46]. Since the patients enrolled in this clinical trial remained symptomatic, we did not withdraw therapy to repeat the measure of BHR but measured BHR while on high-dose therapy to assess for the presence of residual BHR. Studies of occupational exposures and asthma suggest that airway or bronchial hyper-responsiveness can be found in approximately 20% of symptomatic individuals with rates around 10–20% described in general populations; rates vary by race, gender, and atopy [61,62,63]. Use of controller medications can also blunt the response of methacholine [64,65]. We identified a high rate of residual BHR in the consistent with a history of irritant-induced symptoms in both groups. We did not elicit a significant difference in rates of BHR between Uncontrolled and Controlled, most likely because our groups were small.

Chronic cough, paradoxical vocal fold movement, and globus pharyngeus are all laryngeal syndromes that may be due to a common sensory laryngeal dysfunction with overlap in each of the syndromes, and the term “laryngeal hypersensitivity” has been used [66]. We identified high rates of laryngeal hypersensitivity measured by PVFM when provoked with an irritant. This finding did not differentiate between the two groups. Since all patients had irritant exposure, the possibility exists that this exposure may be associated with higher rates of laryngeal hypersensitivity, or the groups may have been too small.

Importantly, BHR and PVFM may co-exist and we hypothesized that irritant-induced hyper-reactivity would be identified as both laryngeal symptoms and BHR. Few patients had overlap of these conditions. However, when we analyzed the presence of either of these measures, or the combination of these measures as a risk for Uncontrolled or Controlled status, we identified a trend, with the presence of BHR and PVFM predicting Controlled status. This finding was contrary to our original hypothesis and suggests that those with residual airway hyper-responsiveness (BHR or PVFM) continue to have the ability to respond to standard asthma therapy, whereas alternative treatments or explanations should be sought in those who no longer display residual airway hyper-responsiveness.

We defined our study population to include those with normal spirometry values. We and others have previously reported the association of small airway dysfunction with symptoms and exposures in many WTC-exposed community members with preserved spirometry [16,35,43,44,67] and we therefore hypothesized that persistent respiratory symptoms would be associated with increased small airway dysfunction in this study. The median values for all measures of forced oscillation, including those considered to assess small airway function (R5–20), were slightly elevated (abnormal) in the group as a whole. There was a suggestion these measures were elevated in Uncontrolled participants, but this finding did not reach statistical significance. Our study groups may have been too small to allow us to reach significance and importantly, all patients had a history of LRS symptoms prior to enrollment in the study, suggesting the possibility of underlying abnormalities in all of the patients. Our finding suggests the need for further study.

Asthma is recognized as a heterogeneous disease that includes multiple inflammatory pathways [47,68], with inflammatory cells that can include neutrophils as well as eosinophils. The absence of peripheral blood Type 2 (T2) inflammatory signals in most of our patients is similar to the findings of Malo et al., who reported on long-term findings in a small group of subjects with irritant-induced asthma [69]. Our findings are also consistent with studies of WTC community members in which low FeNO values or IgE were identified, or in which elevated levels were identified in only a subset of patients with persistent respiratory symptoms [14,16,48,54,55]. The findings are also consistent with studies in firefighters, in which only a subset displayed persistently elevated eosinophil levels associated with decline in lung function [49]. The possibility exists that the uncontrolled LRS symptoms in our patients are associated with neutrophilic pathways of inflammation, which are often less responsive to corticosteroids. We examined blood levels of eosinophils and neutrophils and were unable to identify an association with blood markers and Uncontrolled symptoms. Blood transcript profiling with more detailed analyses might be informative in future studies [70]. Alternatively, blood markers may not be the optimal surrogate for characterizing inflammation within the lung; sputum and biopsy specimens remain the gold standard [51]. Finally, additional mechanisms, including those involving epigenetic changes as a result of exposure, may contribute to respiratory symptoms [71,72,73] and recent studies now suggest the potential for this possibility in WTC-exposed individuals [74,75]. Since, over years, many of these patients improved but did not resolve their LRS, the possibility also exists that the mechanism for these symptoms changed over time from a potentially corticosteroid-sensitive process to one that is no longer sensitive. Our findings reinforce the need for the measurement of additional biomarkers as tools in the continued management of these patients with longstanding WTC-related irritant-induced symptoms.

There are limitations to this study. Studies in diverse populations with individuals who are not “professional” study patients are notoriously difficult to recruit for and we had difficulty with recruitment due to issues with time commitment in working patients, language and literacy issues, and disinterest. As such, there may be bias in our recruitment of the study population. All patients had experienced WTC exposures and reported symptoms prior to enrollment, an inclusion criterion that might also mask some findings. Importantly, despite a high dose of medication and a high rate of adherence, few of our patients obtained Controlled status, and as such we had a small Controlled group for comparison. To enhance our ability to perform comparisons with a group of patients who were controlled at the end of the study, we also modified our study to recruit patients who were controlled before entering the study to enrich the Controlled study population. Analyses of this larger population of those who were controlled or uncontrolled upon entry into the study (n = 73) yielded results that were similar for all parameters presented (data are shown in the Supplement.) The consistency of findings in both groups analyzed reinforced our findings. Despite this additional analysis, the small study group may have reduced our ability to identify statistically significant findings and reinforces the need for further study. We used blood biomarkers and FeNO to assess the inflammatory status of the patients. We, and others, have shown that blood biomarkers, including those for eosinophils, may be variable [76] and FeNO varies over time and control status.

In summary, community members with WTC exposure and persistent uncontrolled lower respiratory symptoms despite treatment with high-dose ICS/LABA may have multifactorial reasons for these symptoms. Importantly, symptom reports consistent with upper airway disease consistently differed between the two groups, a finding with implications for treatment. Moreover, persistent airway hyper-responsiveness as shown by measures of BHR and PVFM was associated with improved control. We identified a suggestion of the presence of continued small airway abnormalities, a finding suggesting further evaluation. This study reinforces the need for continued evaluation of multiple conditions in patients with persistent LRS to help direct appropriate treatment and reinforces the need to focus on upper airway findings in these patients.

## 5. Conclusions

We report results of a study of uncontrolled lower respiratory symptoms in a group of patients treated with standard asthma management with inhaled corticosteroids and long acting beta agonists. Our predominant finding was the difficulty in obtaining control in these patients despite adherence to medications. The association of upper airway symptoms in the group reporting continued symptoms and our finding of persistent airway hyper-responsiveness in those with improved control have important diagnostic and treatment implications and reinforce the need to re-evaluate treatment in these patients.

## Figures and Tables

**Figure 1 ijerph-17-06645-f001:**
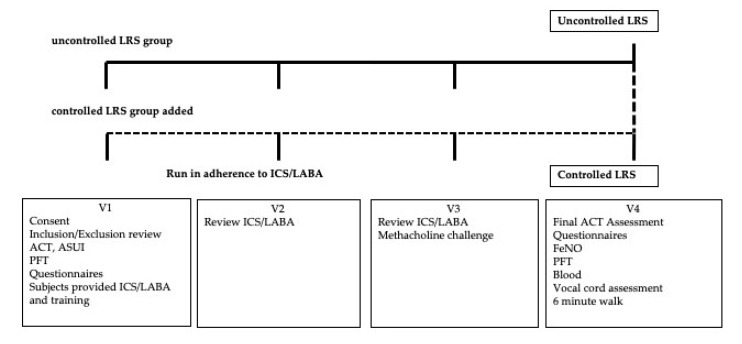
Protocol scheme for study design.

**Table 1 ijerph-17-06645-t001:** Demographic, clinical, and exposure characteristics of patients who were uncontrolled at V1 (n = 60).

	Level	Total	Uncontrolled	Controlled	*p*
n		60	49	11	
Sex, n (%)	F	42 (70)	32 (65)	10 (91)	0.15
	M	18 (30)	17 (35)	1 (9)	
Age, median [IQR]		56.5 [47.8, 62]	57.0 [48.0, 62.0]	56.0 [47.0, 62.0]	0.92
Race/Ethnicity, n (%)	Hispanic	30 (50)	26 (53)	4 (36)	0.04
	NH-White	11 (18)	10 (20)	1 (9)	
	NH-Black	15 (25)	12 (24)	3 (27)	
	Asian and Other	4 (7)	1 (2)	3 (27)	
BMI. median [IQR]		29.4 [26.7, 33]	29.5 [27.1, 32.9]	27.8 [26.0, 34.4]	0.58
BMI, n (%)	Normal (<25)	6 (10)	4 (8)	2 (18)	0.57
	Overweight (25-30)	27 (45)	22 (45)	5 (45)	
	Obese (≥30)	27 (45)	23 (47)	4 (36)	
Dust cloud, n (%)	No	28 (47)	21 (43)	7 (64)	0.32
	Yes	32 (53)	28 (57)	4 (36)	
Exposure category, n (%)	Clean-up Worker	14 (23)	11 (22)	3 (27)	0.36
	Other	5 (8)	3 (6)	2 (18)	
	Resident	7 (12)	7 (14)	0 (0)	
	Local Worker	34 (57)	28 (57)	6 (55)	
ASUI Score, median [IQR]		0.6 [0.5, 0.8]	0.6 [0.4, 0.7]	0.8 [0.7, 0.9]	0.001
MMRC, n (%)	(<3)	49 (82)	39 (80)	10 (91)	0.44
	(≥3)	11 (18)	10 (20)	1 (9)	
6MWT, Median distance, [IQR]		388.8 [345.3, 447.7]	392.7 [344.4, 452.9]	369.0 [358.9, 434.0]	>0.99

**Table 2 ijerph-17-06645-t002:** Upper airway symptoms in Uncontrolled and Controlled study participants (n = 60).

	Total	Uncontrolled	Controlled	*p* Value
ICSD, median [IQR]	31.0 [19.0, 41.0]	35.0 [24.0, 42.5]	26.0 [18.2, 29.5]	0.027
LCQ, median [IQR]	12.7 [9.0, 16.2]	12.0 [8.9, 15.4]	15.6 [12.7, 20.2]	0.047
VHI, median [ IQR]	7.0 [0.5, 16.5]	7.0 [1.0, 17.0]	1.0 [0.0, 9.8]	0.18

**Table 3 ijerph-17-06645-t003:** Spirometry and forced oscillation measures in patients who were uncontrolled at V1 (n = 60).

	Level	Total	Uncontrolled	Controlled	*p*-Value for % Pred
**SPIROMETRY**					
Pre BD					
FVC,L (% of predicted, median)		3.2 (98)	3.2 (97)	3.3 (100)	0.59
FEV_1_,L (% of predicted, median)		2.5 (93)	2.5 (92)	2.3 (96.5)	0.24
FEV_1_/FVC, median		78.5 [73.2, 82.4]	78.5 [72.4, 82.4]	81.2 [75.3,83.4]	0.34
Post BD					
FVC,L (% of pred, median)		3.1 (99)	3.2 (98.5)	3.1 (100)	0.57
FEV_1_,L (% of pred, median)		2.5 (96)	2.6 (96)	2.4 (96.5)	0.81
FEV_1_/FVC, median		78.3 [75.1, 82.3]	78.3 [75.0, 82.2]	77.7 [75.4,81.1]	0.98
FOT Measurements					
Pre BD					
R5 median [IQR]		4.3 [3.4, 5.7]	4.4 [3.4,5.7]	3.7 [3.3, 5.7]	0.82
R5–20 median [IQR]		0.9 [0.4, 1.5]	1.0 [0.5,1.5]	0.6 [0.4, 1.4]	0.62
Post BD					
R5 median [IQR]		4.1 [3.4, 5.4]	4.1 [3.5,5.3]	4.0 [3.4, 5.7]	0.84
R5–20 median [IQR]		0.8 [0.4, 1.2]	0.8 [0.4,1.2]	0.6 [0.4, 1.1]	0.73

**Table 4 ijerph-17-06645-t004:** Airway hyper-responsiveness measured by bronchial hyper-reactivity (BHR) and paradoxical vocal fold movement (PVFM) in Uncontrolled and Controlled participants (n = 60).

	Level	Total	*n* (% Controlled)	*p* Value
BHR (at dose ≤16mg)	*Negative*	24	3 (12.5)	0.46 *
	*Positive*	24	6 (25)	
	*NA*	12		
Any PVFM	*Negative*	29	5 (17)	0.74 *
	*Positive*	27	6 (22)	
	*NA*	4		
PVFM unprovoked	*Negative*	50	10 (20)	>0.99 *
	*Positive*	5	1 (20)	
	*NA*	5		
PVFM provoked	*Negative*	29	5 (17)	0.74 *
	*Positive*	27	6 (22)	
	*NA*	4		
Any PVFM and BHR	*-* *PVFM, -BHR*	9	0 (0)	0.02 **
	*-PVFM, +BHR*	14	3 (21)	
	*+PVFM, -BHR*	14	3 (21)	
	*+PVFM,+BHR*	9	3 (33)	
	*NA*	14		

* Fisher exact test; ** exact Cochran–Armitage trend test (catt_exact).

**Table 5 ijerph-17-06645-t005:** Inflammatory markers in Uncontrolled and Controlled participants.

	Total	Uncontrolled	Controlled	*p*
n	60	49	11	
FeNO, ppb median [IQR]	16.5 [13.1, 21.8]	17.8 [14.0, 22.2]	13.0 [10.2, 13.9]	0.01
Blood markers				
White blood cells *, Median [IQR]	7.2 [5.2, 8.5]	7.1 [5.0, 8.6]	7.2 [6.5, 8.1]	0.70
Eosinophils *, Median [IQR]	0.10 [0.10, 0.20]	0.10 [0.10, 0.20]	0.10 [0.10, 0.10]	0.34
Neutrophils *, Median [IQR]	3.8 [3.0, 5.1]	3.7 [2.7, 5.1]	4.1 [3.7, 4.8]	0.37
Lymphocytes *, Median [IQR]	2.0 [1.6, 2.8]	2.0 [1.5, 2.9]	2.2 [1.8, 2.5]	0.65
ELR, Median [IQR]	0.06 [0.04, 0.11]	0.1 [0.0, 0.1]	0.1 [0.0, 0.1]	0.39
ENR, Median [IQR]	0.03 [0.02, 0.06]	0.0 [0.0, 0.1]	0.0 [0.0, 0.0]	0.18
NLR, Median [IQR]	1.78 [1.30, 2.47]	1.8 [1.2, 2.5]	1.8 [1.5, 2.3]	0.76
Total IgE, median [IQR]	37 [19.8, 115.5]	41.0 [20.0, 135.0]	31.0 [21.5, 57.0]	0.52
Any allergen-specific IgE, n (%)	26 (43.8)	23 (47)	3 (27)	0.32

*: (×
10^9^/L).

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
