# Peer review of "Characterization of Persistent Uncontrolled Asthma Symptoms in Community Members Exposed to World Trade Center Dust and Fumes"

_ijerph, 2020, doi:10.3390/ijerph17186645_

Round 1

Reviewer 1 Report

In this study, the authors identify a set of conditions associated with uncontrolled lower respiratory symptoms (LRS) from a patient cohort that uses inhaled corticosteroid treatment. The comprehensive background provided in this study, combined with the study design and results, are a good fit for this journal, but could be strengthened with the following additions:

Major:

  • Section 2.3/2.4: Increased neutrophil counts are usually associated with corticosteroid resistant asthma (See PMCID: PMC5711587).
    • The authors should include neutrophil counts, since that metric may explain low FeNO values.
    • Eosinophil/lymphocyte ratios and eosinophil/neutrophil ratios, along with neutrophil/lymphocyte ratios should be included in the metrics to provide more granularity into characterizing the inflammatory responses between controlled and uncontrolled groups.

Minor:

  • Section2.1/2.2: It would help the reader to provide some more detail regarding the one-arm intervention study characteristics, along with the rationale for using this type of study. Additional information with regards to the Asthma Control Test and a diagram/workflow of the three-month plan would also be beneficial here. 
  • Section 3.6: Inflammation plays a significant role across asthma phenotypes (See PMCID: PMC5901974, PMC7160128, PMC7344461). The associated values in table 1 (Total IgE, Allergen IgE, Eosinophil counts) should be included in a separate table.
  • Section 3: From an organizational standpoint , re-ordering the results sections would provide a clearer overview of the significant findings, which were a bit challenging to identify. Sections with Table 1 should precede sections with Table 2, for example. 
  • Section 4: The authors provide a comprehensive summary of their findings, but do not discuss epigenetic mechanisms as potential reasons for differences between the uncontrolled and controlled groups (See PMCID: PMC5472054, PMC3775260). Chronic exposures and epigenetic priming should be included in the discussion. 
  • Captions should be included with every table, indicating the metric(s) measured between/across groups, and the statistical test used to generate the p-value as indicated in the final column of each table. 

Author Response

Response to Reviewer 1

We would like to thank the reviewers for their detailed and careful review of our manuscript entitled “Characterization of Persistent Uncontrolled Asthma Symptoms in Community Members Exposed to World Trade Center Dust and Fumes,” by Reibman et al. (ID ijerph-901260). In response to their substantive critiques, we have significantly reorganized the manuscript and revised the analyses. We believe that these revisions make this manuscript much stronger, and we are very appreciative of their comments. We believe that our revised manuscript is of value and pertinent to this edition of the journal, which focuses on the adverse health effects of WTC exposures. Moreover, we believe this manuscript is of clinical relevance. We provide a point by point response to the reviewers. 

Reviewer 1

Major comments

Reviewer 1 points out the importance of neutrophils as an endo-phenotype associated with uncontrolled asthma and requested that we add neutrophil counts, and eosinophil ratios with other inflammatory cells to add more granularity to characterizing the inflammatory responses between controlled and uncontrolled groups.

We have now added a statement in our introduction (Line 78) that we analyzed blood inflammatory markers. As recommended by the reviewers, we have also added a new section (Line 328) and a new Table 5, in which we show accessible blood biomarkers of inflammation, including neutrophil counts, eosinophil/lymphocyte ratios, eosinophil/neutrophil ratios, and eosinophil/lymphocyte ratios. We discuss these findings now in the Discussion (Line 450 with new references)

Minor comments

We have added additional detail regarding the one-arm intervention study characteristics and rationale on Line 87 and a new reference.

We have added additional information about the Asthma Control Test (Line 385)

We have now included a new figure with a diagram of the three-month plan Fig. 1.

As suggested by the reviewer, we have now added a new Table 5 showing accessible blood inflammatory biomarkers

We have re-ordered the tables and removed sections that are now shown in Table 5.

The reviewer would like a discussion of epigenetic findings in the discussion and we have added a brief statement about epigenetic priming and asthma and WTC findings in the discussion (Line 465)

We have now added captions for each table and the metrics measured across groups are described more clearly in the text (eg Line 166, 276). We have discussed the statistical tests in more detail in the Methods section (e.g. Line 194).

Reviewer 2 Report

Major points

1. Are there any papers that describe the difference between Controlled and Uncontrolled among ''usual'' asthma in terms of parameters used in this manuscript?   2. No pathophysiological difference was found in this study. As subgroup  analysis, an analysis between Controlled (ACT>=20) and Uncontrolled (ACT<19) or (ACT<18) might be able to find factors that are causing the symptoms.   3. A: The data of normal range of the factors in Table1-4 would be very helpful to interpret the result.     B: The comparison of the data of Uncontrolled and the 'normal' would find factors that are causing the symptoms, although that is not the original purpose of study.  

Author Response

Response to reviewers

We would like to thank the reviewers for their detailed and careful review of our manuscript entitled “Characterization of Persistent Uncontrolled Asthma Symptoms in Community Members Exposed to World Trade Center Dust and Fumes,” by Reibman et al. (ID ijerph-901260). In response to their substantive critiques, we have significantly reorganized the manuscript and revised the analyses. We believe that these revisions make this manuscript much stronger, and we are very appreciative of their comments. We believe that our revised manuscript is of value and pertinent to this edition of the journal, which focuses on the adverse health effects of WTC exposures. Moreover, we believe this manuscript is of clinical relevance. We provide a point by point response to the reviewers.

Reviewer 2

Major points

  1. The reviewer would like additional information about Controlled and Uncontrolled patients among “usual” asthma patients.

We have included additional information on Lines 385 about the use of the ACT.

  1. The reviewer requested a subgroup analysis of patients who had an ACT between 18 and 19.

We agree with the reviewer that this is an important question about the degree of control. The ACT has been shown to discriminate between control with a cut off of < 15 consistent with poorly controlled asthma (Nathan et al, Schatz et al.). As such, to answer the reviewers’ question, we reanalyzed the data comparing those who were very Uncontrolled (ACT score < 15, n= 29) with Controlled (n = 19). We did not detect a significant difference in basic characteristics, lung function (spirometry or FOT) (data discussed Line 281, data shown in Supplement S3a), AHR, or inflammatory markers. We continued to find a significant difference between upper airway symptoms (stated, but data not shown Line 260). We include statements to this effect within each paragraph.

As requested, we have included the statement of % of predicted and clarified the normal range for the FOT values on Line 166, 276).

Reviewer 3 Report

The submitted article concerned examination of asthma symptoms in 9/11 survivors. Unfortunately, There were too many inconsistencies in methodology and reporting of results for it to be acceptable in the current form. My biggest concern is that the authors changed the design (inclusion criteria), based on interim analyses, midway through the project when they didn't see the expected results. After completing the study, the authors spent a lot of time laying out results (in both tables and text), and the vast majority of the data indicated no differences between the groups. The phrase "the data suggests ..." was used more than once, but no significance was found. I realize the concern with statistical significance per se, i.e., but not a single effect size was noted. The authors examined a number of DVs and basically there weren't any differences between the groups. The primary reason that I didn't call for outright rejection was that, in my opinion, the authors stumbled upon something interesting. The prevailing thought had been that those with "uncontrolled" symptoms failed to reach control because they hadn't adhered to medication protocol, and the authors determined that this was NOT the case - so in the few instances where something was significant, it was not because of that difference. If the paper can be rewritten to focus on this finding it would add to the literature in my opinion. First, mention the prevailing thought on an inability by some to control their asthma. Next, simply examine the "uncontrolled" group pre- and post-medication regimen and note that, although medication protocol WAS followed, no differences in the measures they examined were found. I don't recall seeing exactly (the authors state "a few") how many of those DID reach control, and realize that the likely small number will prevent significance, so look at effect sizes too. Maybe something will show up. If not, unfortunately, at that point I'd vote for a rejection for publication.

Author Response

Response to reviewers

We would like to thank the reviewers for their detailed and careful review of our manuscript entitled “Characterization of Persistent Uncontrolled Asthma Symptoms in Community Members Exposed to World Trade Center Dust and Fumes,” by Reibman et al. (ID ijerph-901260). In response to their substantive critiques, we have significantly reorganized the manuscript and revised the analyses. We believe that these revisions make this manuscript much stronger, and we are very appreciative of their comments. We believe that our revised manuscript is of value and pertinent to this edition of the journal, which focuses on the adverse health effects of WTC exposures. Moreover, we believe this manuscript is of clinical relevance. We provide a point by point response to the reviewers and since reviewer 3 had comments that required a major re-analysis and presentation of our data, we will start with the response to this reviewer.

Reviewer 3

Major points

The reviewer was most concerned with the change in the design based on our interim analysis.

The reviewer was concerned about the absence in differences between the two groups and the use of the phrase “the data suggest” when no significance was found and that no effect size was noted.

The reviewer suggested a refocus of the manuscript on the finding that the prevailing thought is that an inability to control asthma is due to a failure of adherence. Then the reviewer would like an examination of the “Uncontrolled” group pre and post medication regimen and how many of the participants did reach control.

We thank the reviewer for suggesting these changes and believe that they are completely appropriate and improve the manuscript. As such, we have reanalyzed all our data using the originally planned group that began the study as uncontrolled (n = 60). Of this group, only 11 (18%) gained Controlled status by V4 and 49 remained Uncontrolled. We clarify this change in analysis in the Methods (101). We now use this group for our comparison as we had planned in the original design of the study and all tables in the manuscript report on this group. We describe this revised analysis in our Abstract, at the beginning of our Results section (Line 203) and have modified all tables to present data from this group (n = 60 subjects who began the study as uncontrolled). We state that few patients reached Controlled status in our first results on in the results (Line 203) and in the discussion and point out that this necessitated an alternative search for explanations for their symptoms (Lines 368, Lines 380).

We also present the data from the whole group, in which we modified our recruitment and recruited 73 individuals, in the supplemental tables in the Appendix. The analysis of the whole group supports the findings in our original study group, reinforcing the strength of our analyses. We agree with the reviewer that a refocus of the manuscript to describe that few participants reached “Controlled” is important and makes the manuscript clearer. We are very appreciative of the suggestion to refocus the manuscript and the analyses.

Our findings remain the same as those presented in our original submission. Using this smaller but cleaner study group, we continue to identify a significant difference in upper airway symptoms between those that are Uncontrolled and those that are Controlled. We believe that this is an important finding since it has major implications for treatment – ie. high dose ICS may not be needed in these patients, and a greater focus should be made to treat these upper airway symptoms.

In addition, our findings in this original group, as well as in the larger group, suggest high rates of airway hyperreactivity shown by measurements of bronchial hyperreactivity and PVFM. Additional new analyses showed a trend towards the presence of persistent airway reactivity, measured as bronchial hyperreactivity or paradoxical vocal fold movement, and Controlled status. This is a new finding that is now shown in the Results section (Line 286, 299, 311) and is shown in a new Table 4. It is also replicated in our appendix where we analyze the larger group. We believe that this is an important finding that suggests that the persistence of airway hyperreactivity is associated with improved response to treatment. We have added this to the discussion (Line 405).

Round 2

Reviewer 2 Report

I think that the manuscript is fine for the publication.

Reviewer 3 Report

The authors have submitted a much improved manuscript concerning asthma symptoms in WTC survivors. The re-analysis of the data was well-done. I believe it's a much stronger paper - the results are unexpected given past hypotheses, but that makes it more interesting.